# Evaluating the Therapeutic Efficacy of an Anti-BAFF Receptor Antibody Using a Rheumatoid Arthritis Mouse Model

**DOI:** 10.3390/antib14040090

**Published:** 2025-10-20

**Authors:** Adi Aharon, Rachel Birnboim-Perach, Omer Grotto, Adi Amir, Daniel Diadko, Nitzan Beltran, Limor Nahary, Itai Benhar

**Affiliations:** 1Department of Molecular Microbiology and Biotechnology, The Shmunis School of Biomedicine and Cancer Research, The George S. Wise Faculty of Life Sciences, Tel-Aviv University, Tel-Aviv 6997801, Israel; adidiaharon@gmail.com (A.A.); rachelibir@gmail.com (R.B.-P.); omer.grotto@gmail.com (O.G.); adi.amir93@gmail.com (A.A.); danield2601@gmail.com (D.D.); naharyl@yahoo.com (L.N.); 2Department of Pathology, Wolfson Medical Center, Holon 5822012, Israel; nitzan2t@gmail.com; 3Cancer Biology Research Center, Tel Aviv University, Tel-Aviv 6997801, Israel

**Keywords:** rheumatoid arthritis (RA), BAFF receptor (BAFF-R), neutralizing antibody, collagen-induced arthritis (CIA) mouse model

## Abstract

Background: Rheumatoid arthritis (RA) is a chronic autoimmune disease characterized by joint inflammation that leads to tissue damage and disability. RA affects approximately 0.5–1% of the global population and is driven by a complex interplay of genetic susceptibility, environmental factors, and immune dysregulation. While biologic and targeted synthetic DMARDs improved RA treatment, they have limitations in efficacy, safety, and accessibility. B-cell-targeting therapies, such as anti-CD20, have shown effectiveness, but only with broad immunosuppression, which can increase infection risk and compromise humoral immunity. Therefore, there is an unmet need for more selective therapeutic strategies that modulate pathogenic immune pathways while preserving protective immune functions. It has been suggested that targeting the BAFF pathway may offer a more favorable therapeutic approach compared to targeting CD20. Objectives: In this study, we evaluated the therapeutic potential of V3-46s mIgG2a, an anti-BAFF-R (BR3) antibody in a mouse RA model, hypothesizing that it would offer a more selective and effective strategy. Methods: We expressed and purified four antibody variants and assessed their binding and neutralizing activity in vitro. V3-46s mIgG2a was selected for in vivo evaluation in a collagen-induced arthritis (CIA) model. Results: Treatment with this antibody delayed disease onset and reduced arthritis severity, spleen index, and B-cell populations. Conclusions: These findings highlight the potential of BAFF-R-targeting antibodies as a therapeutic approach for RA treatment. This preclinical work lays the groundwork for future development of BAFF-R blockade as a complementary or alternative strategy to current biologic treatments.

## 1. Introduction

Rheumatoid arthritis (RA) is a chronic systemic autoimmune disease characterized by persistent synovial inflammation, joint destruction, and extra-articular manifestations [1]. Affecting approximately 0.5–1% of the global population, RA predominantly targets the small joints of the hands and feet, causing pain, stiffness, swelling, and progressive disability [2]. The pathogenesis of RA involves a complex interplay between genetic susceptibility—particularly HLA-DRB1 alleles—and environmental triggers, including smoking, urbanization, and hormonal factors [3].

The hallmark of RA is pannus formation, a hyperplastic and invasive synovial tissue that erodes cartilage and bone. This inflammatory process is driven by the activation of both innate and adaptive immune responses. Key immune effectors include dendritic cells, macrophages, neutrophils, and fibroblast-like synoviocytes, as well as autoreactive T and B lymphocytes. These cells interact in a proinflammatory milieu enriched with cytokines such as tumor necrosis factor-alpha (TNF-α), interleukin-1 (IL-1), and interleukin-6 (IL-6), which perpetuate the immune response and tissue destruction [4].

RA patients are initially offered treatments involving disease-modifying antirheumatic drugs (DMARDS), such as methotrexate (MTX), which is a first-line treatment for RA [5]. However, when these fail, patients are switched to treatment using biologics [6]. B-cells play a critical role in RA by producing autoantibodies such as rheumatoid factor (RF) and anti-cyclic citrullinated peptide (anti-CCP), promoting the formation of immune complexes and complement activation [7]. In addition, B-cells contribute to RA pathogenesis through cytokine secretion and antigen presentation to T cells, sustaining the inflammatory response [8].

Given their role in disease progression, B-cells have become important targets in RA therapy. One established approach involves CD20-targeting monoclonal antibodies, such as rituximab, which deplete CD20+ B-cells from the circulation [9]. While effective in many patients, CD20 is broadly expressed on most mature B-cells, including regulatory and non-autoreactive B-cells. This broad depletion may result in increased risk of infections and impaired humoral immunity [10]. This excellent recent review provides a thorough list of potential therapeutic approaches for depleting B-cells in RA.

B-cell-activating factor (BAFF), a TNF superfamily cytokine, is essential for B-cell maturation and survival via its interaction with BAFF-R (BR3). In contrast to CD20, BAFF-R (B-cell-activating factor receptor, BR3) is more selectively expressed on transitional and mature B-cells, with elevated expression on autoreactive B-cell subsets implicated in autoimmune diseases. Perhaps more relevant for therapeutic purposes, elevated levels of BAFF and BAFF-R expression have been observed in many autoimmune conditions, suggesting that targeting this pathway may be warranted in such instances [11,12]. RA is among the autoimmune conditions that were defined as such, as elevated levels of BAFF and BAFF-R expression found in RA cells [13], and were reported to correlate with disease severity in RA patients, leading to studies of blocking the BAFF/BAFF receptors pathway as potential therapeutic approaches for RA [14,15,16]. Antibodies that target BAFF (such as Belimumab) have been suggested for treating RA but have not yet been FDA-approved for this indication [17,18]. While BAFF can engage multiple receptors (BAFF-R, TACI, and BCMA), BAFF-R shows notably stronger expression than the other receptors, specifically in autoreactive B-cell populations [19]. Therefore, targeting BAFF-R could enable a more specific modulation of the B-cell compartment, sparing beneficial B-cell populations while reducing pathogenic ones. Thus, anti-BAFF-R antibodies may offer a more refined therapeutic approach with fewer off-target effects than CD20-directed therapies (perhaps because anti-CD20 therapy (e.g., rituximab) effectively depletes circulating naive and memory B-cells but leaves marginal zone and plasma cells less affected); BAFF-R targeting may also be combined with anti-CD20 antibodies for more effective therapy of RA [18,20].

The collagen-induced arthritis (CIA) mouse model is widely used for studying the pathophysiology of RA and evaluating potential treatments. It mimics key features of human RA, including synovial inflammation, pannus formation, cartilage destruction, and bone erosion. CIA is typically induced in genetically susceptible mouse strains (such as DBA/1) via immunization with type II collagen emulsified in complete Freund’s adjuvant, followed by a booster with incomplete Freund’s adjuvant. The immune response generated against collagen leads to autoimmune joint inflammation, providing a robust and reproducible platform for preclinical testing [21]. The severity of arthritis is quantified through clinical scoring and histopathological analysis, including assessment of inflammatory infiltrates, synovial hyperplasia, and joint damage [22].

In this study, we aimed to evaluate the therapeutic potential of a monoclonal antibody targeting BAFF-R in the CIA mouse model. By inhibiting the BAFF/BAFF-R axis, which is crucial for B-cell survival and activation, this therapeutic strategy may attenuate disease progression more selectively than traditional B-cell depletion approaches. This work seeks to provide foundational evidence for the utility of BAFF-R targeting in RA and antibody isotype, to inform the development of more precise immunotherapies for patients unresponsive to current treatments.

## 2. Materials and Methods

### 2.1. Production of Anti-BAFF-R (BR3) Antibodies

A synthetic gene encoding the variable domains of an anti-BR3 antibody was designed according to sequences taken from patent no. US2010O280227A1 [23], obtained from IDT and cloned into pcDNA3.4-IgH and pcDNA3.4-IgK vectors [24]. These plasmids are based on the CMV promoter-driven pcDNA3.4 backbone included in the Thermo Fisher Expi293TM kit for transient transfection-based expression. We used plasmids that contained the constant regions of either a murine IgG2a (efficient in recruiting immune effector functions) or a murine IgG1 antibody (deficient in recruiting immune effector functions) [25]. This resulted in the following plasmids: pcDNA3.4-V3-46s murine IgG2a, used for the expression of V3-46s mIgG2a antibody; pcDNA3.4-V3-46s murine IgG1, used for the expression of V3-46s antibody mIgG1; pcDNA3.4-V3-1 murine IgG2a, used for the expression of V3-1 mIgG2a antibody; and pcDNA3.4-V3-1 murine IgG1, used for the expression of V3-1 antibody mIgG1.

The anti-BR3 monoclonal antibodies were expressed by transient transfection of Expi293F™ cells using the ExpiFectamine™ 293 transfection kit (Gibco, Weltham, MA, USA) in 30 mL cultures. Heavy and light chain expression plasmids were co-transfected at a 1:3 ratio (7.5 μg IgH: 22.5 μg IgL) following the manufacturer’s protocol. On day 7 post-transfection, supernatants were harvested by centrifugation (8500 rpm, 10 min, 4 °C), buffered with 20 mM phosphate buffer (pH 7.0), and filtered through a 0.45 μm syringe filter (Millipore, Burlington, MA USA). Murine IgG2a antibodies were purified using Protein A affinity chromatography (MabSelect™ SuRe™, Cytiva, Marlborough, MA USA): columns were equilibrated with PBS, eluted with 0.1 M citric acid (pH 3.0) to release bound antibodies in 1 mL fractions, and immediately neutralized with 200 μL of 1.5 M Tris-HCl (pH 8.8), while murine IgG1 antibodies were purified using Protein G affinity chromatography (Cytiva, Marlborough, MA, USA): columns were equilibrated with PBS, eluted with 0.1 M glycine (pH 2.7) to release bound antibodies in 1 mL fractions, and immediately neutralized with 60 μL of 1.5 M Tris-HCl (pH 8.8). Fractions were analyzed by SDS-PAGE, pooled based on purity, and buffer-exchanged into PBS using PD-10 columns (Cytiva, Marlborough, MA, USA). Purified antibodies were aliquoted and stored at −80 °C.

### 2.2. Evaluation of Anti-BR3 Antibodies’ Binding Abilities Using ELISA and Flow Cytometry

The binding capacity of anti-BR3 antibodies was evaluated by both indirect ELISA and flow cytometry. For ELISA, 96-well plates (Nunc, Stockholm, Sweden) were coated overnight at 4 °C with 5 µg/mL recombinant mBR3-rFc (Biointron, Shanghai, China) or rabbit IgG (negative control antigen), then blocked with 3% (*w*/*v*) skim milk in PBS (MPBS) for 1 h at 37 °C. After three washes with PBS containing 0.05% Tween 20 (PBST), serial 3-fold dilutions of anti-BR3 antibodies were added and incubated for 1 h at 37 °C. HRP-conjugated goat anti-mouse (H+L) secondary antibody (Jackson ImmunoResearch Labs, USA, West Grove, PA, USA, Cat: 115-035-062) was applied for 1 h at room temperature, and signal was developed using TMB substrate (Dakocytomation, Carpinteria, CA, USA) and stopped with 1 M H_2_SO_4_. Absorbance was measured at 450 nm using an EMax Plus microplate reader (Molecular Devices, San Jose, CA, USA). For flow cytometry, 0.6 × 10^6^ i29 cells [26] or 1.4 × 10^6^ mouse splenocytes were washed with 1% bovine serum albumin (Sigma (Merck). Saint Louis MO, USA) in PBS (PBS/BSA), blocked with 3% PBS/BSA, and treated with murine Fc block (miltenyi biotech, Bergisch Gladbach, Germany). Cells were incubated with 5 µg/mL V3-46s mIgG2a or isotype control (Bioxcell, Lebanon, NH, USA, clone C1.18.4 (Cat: BE0085)) for 30 min at room temperature, followed by fluorescently labeled goat anti-mouse secondary antibodies (AF488) (Abcam, UK, Cambridge, UK, ab150117). Dead cells were excluded based on DAPI staining. Acquisition was performed using a CytoFLEX 4L flow cytometer (Beckman Coulter, Brea, CA, USA), and data was analyzed with Kaluza.

### 2.3. Evaluation of the Ability of Anti-BR3 Antibodies to Neutralize BAFF-Induced B-Cell Proliferation In Vitro

Splenocytes from C57BL/6 mice were obtained and 1 × 10^5^ cells per well were seeded (diluted in RPMI supplemented with 10% [*v*/*v*] FBS, PS (Penicillin-Streptomycin), and 0.01 mM β-Mercaptoethanol (Merck, Darmstadt, Germany)). The cells were cultured for 72 h at 37 °C, 5% CO_2_ with various concentrations of soluble anti-BR3 antibodies or isotype control (IC) mAbs (mIgG2a- BioXCell, clone C1.18.4 mIgG1- Bioxcell, clone MOPC-21 cat: AB_1107784) and with or without recombinant mouse BAFF (Biolegend, San Diego, CA, USA, catalog 591202). Cells were then stained for CD45 (Biolegend, catalog BLG-103108), B220 (Miltenyi, catalog 130-110-846), CD21/CD35 (Miltenyi, catalog 130-111-732), CD23 (Miltenyi, catalog 130-118-761), and DAPI (Sigma, catalog D9542), and were visualized on a CytoflexL4 (Beckman Coulter) flow cytometer. The data was analyzed using Kaluza analysis software version 2.1.1 (Beckman Coulter).

### 2.4. Mouse Model of Collagen-Induced Arthritis

Male, 7–8-week-old DBA/1 mice (a mouse RA model that closely mimics the human condition and where BAFF was shown to be involved [27,28]) were obtained from Envigo, Horst, Netherlands. All animal experiments were conducted in accordance with the guidelines approved by the Institutional Animal Care and Use Committee of Tel Aviv University (approval number: TAU-LS-IL-2205-158-5). Arthritis was induced by an intradermal injection at the base of the tail using an emulsion of immunization-grade Bovine Type II Collagen (CII, Chondrex, Inc. Woodinville, WA, USA) in complete Freund’s adjuvant (CFA, Sigma) containing Mycobacterium tuberculosis H37 RA at 2 mg/mL (BD, Franklin Lanes, NJ, USA) on day 0. On day 21, the mice received a booster injection consisting of incomplete Freund’s adjuvant (IFA, Sigma) and immunization-grade Bovine Type II Collagen, as previously described [21]. Control mice were injected with CFA/IFA and PBS instead of collagen. The treatment groups received either PBS, isotype control murine IgG2a (BioXcell clone C1.18.4), or anti-BR3 antibody V3-46s mIgG2a at a concentration of 6 mg/kg every 2–3 days starting from the day of disease onset (2 days following the booster injection and a total of 5 treatment injections). On day 32, when the PBS and IC mice reached the maximal RA score that we could observe in these experiments, the mice were bled and euthanized, and their paws were collected for histology analysis.

### 2.5. CIA Severity Scoring Method

The severity of CIA was assessed using a validated clinical scoring system, where each animal’s total score was the sum of the scores for all four paws, on a scale of 0–16. Each paw was individually scored as described in Table 1. In some experiments, the number of arthritic paws was quantified, and the total score was then divided by the number of affected paws to provide an average severity score.

### 2.6. Sera Collection

Blood was collected from the facial vein of each mouse on the day of euthanization and then centrifuged at 14,000 RPM at 4 °C in an Eppendorf Microfuge. The supernatant was collected and centrifuged again under the same conditions. The supernatant (containing the sera) was collected and stored at −20 °C. Sera collected were used for titer ELISA (as described above).

### 2.7. Immune Phenotyping

At the end of the experiments, the spleens were harvested, and single-cell suspensions were prepared. The fresh splenocytes were stained using conjugated antibodies. B-cells were defined as CD45+B220+, macrophages were defined as CD45+F4/80+, and neutrophils were defined as CD45+Ly6G+. FITC anti-mouse CD45 was obtained from Biolegend (BLG-103108). Antibody CD11b-APC-Vio770 was obtained from Miltenyi (130-113-803), antibody B220-PE was obtained from Miltenyi (130-110-846), antibody Ly6G-PerCP-Vio700 was obtained from Miltenyi (130-117-500), and antibody F4/80-APC was obtained from Miltenyi (130-116-547). The cells were incubated with the antibodies for 30 min in the dark on ice, and DAPI was used as a marker for cell death. Cell suspensions were then washed and analyzed using Cytoflex flow cytometer (Beckman Coulter) and Kaluza analysis software (Beckman Coulter, Brea, CA, USA).

### 2.8. Processing and Evaluation of Joint Histology

The mice joints were dissected, and the samples were fixed in 4% paraformaldehyde (Bar-Naor, Petah Tikva, Israel) for at least 24 h before being placed in a decalcifier (National Diagnostics, Atlanta, GA, USA). The joints were then embedded in paraffin, and 5-µm sections were prepared for histopathological evaluation. These sections were stained with hematoxylin and eosin (H&E) (Leica ST5020 Multi-stainer, Nussloch, Germany). An experienced pathologist, blinded to the study conditions, examined the tissue sections and scored them for inflammation, pannus formation, bone resorption, and cartilage damage on a scale of 0 to 5.

### 2.9. Statistical Analysis

Statistical significance was assessed using a one-way ANOVA, which is appropriate for comparing the means of more than two independent groups under the assumption of normally distributed data and homogeneity of variances. Post hoc, multiple comparisons were corrected using the Bonferroni method to control for type I error [29]. In cases involving comparisons between two groups, an unpaired two-tailed Student’s T-test was performed [30], using GraphPad Prism 10 (GraphPad Software, Inc. version 10.2.0). Data is shown as mean ± SD or mean ± SEM as indicated. * *p* < 0.05, ** *p* < 0.01, *** *p* < 0.001, and **** *p* < 0.0001.

## 3. Results

### 3.1. Design, Expression, and Purification of Anti-BR3 Antibodies

US patent 2010/0280227 of Genentech discloses antibodies that bind, and some that also block mouse and human BR3 [23]. These antibodies were evaluated using various models of autoimmune conditions, but RA was not one of them. Our research objectives included the evaluation of the potential of such antibodies to alleviate the severity of RA in a mouse model. To do so, two different variable domains of both heavy and light chains taken from [23] were chosen, based on the highest binding affinity and mouse BR3 blocking potency that was reported.

The antibodies were produced in Expi293F™ cells system as mIgG2a and mIgG1 isotypes. To produce the antibodies, each VH or VL was cloned into the pCDNA3.4 plasmid, containing the appropriate constant domain of either heavy or light chain of mIgG2a or mIgG1 isotype [24]. The different antibodies were V3-1, which is the original clone of an antagonistic anti-BR3 antibody, and V3-46s, which is an affinity-matured variant of the same clone. Those antibodies were chosen because of their high affinity to BR3 (both murine and human) and antagonistic activity, meaning BR3-related B-cell proliferation prevention [23].

All four anti-BR3 antibodies were expressed in Expi293F™ cell system, and the conditioned medium collected from the cells 7 days post-transfection was purified on either a MabSelect© or a protein G affinity column (Cytiva). The purification process was analyzed by SDS-PAGE, as shown in Appendix A. The purified proteins were buffer exchanged to PBS by PD-10 chromatography as described in the manufacturer’s protocol. As shown in Appendix A, three of the antibodies (MW:~150KDa) were expressed well, while one (V3-46s mIgG1) showed poor expression. V3-46s mIgG2a with 1.8 mg protein recovered from 30 mL transfection, V3-46s mIgG1 with 0.108 mg protein recovered from 30 mL transfection, V3-1 mIgG2a with 1.4 mg protein recovered from 30 mL transfection, and V3-1 mIgG1 with 1.2 mg protein recovered from 30 mL transfection.

### 3.2. In Vitro Characterization of the Anti-BR3 Antibodies

#### 3.2.1. All the Anti-BR3 Antibodies Bind BR3

All four antibodies were analyzed by ELISA for their binding ability and specificity to BR3-rFc (fusion to rabbit FC), compared to rIgG as a negative control. The antibodies were detected by HRP-anti-mouse IgG (H+L). The results are shown in Figure 1A. According to these results, the apparent affinity of V3-1 mIgG2a to mBR3 is higher than the other three antibodies. Nevertheless, all four antibodies show EC_50_ values of between 0.6 and 2.3 nM, which is a good affinity.

#### 3.2.2. V3-46s mIgG2a Antibody Bind BR3 on i29 Cells and Splenocytes

After evaluating the binding abilities of all four antibodies using ELISA, we performed a cell proliferation assay and chose the V3-46s mIgG2a antibody as our treatment. Therefore, we further evaluated the binding ability of the mAb to BR3 on cells, using both splenocytes isolated from C57BL/6 mice single-cell suspension and i29 cells. As shown in Figure 1B, there is an efficient binding to both i29 cells and to freshly isolated mouse splenocytes in comparison to the isotype control and the secondary antibody only.

### 3.3. V3-46s mIgG2a Exhibits Neutralization of BAFF-Induced B-Cell Proliferation In Vitro

To evaluate the blocking and neutralization activities of the anti-BR3 mAbs, a BAFF-activated B-cell proliferation assay was conducted. Herein, splenocytes (1 × 10^6^ cells/mL) from C57BL/6 mice were incubated for 72 h with/without BAFF [7 ng/mL] and with/without anti-BR3 or IC mAbs [3 ng/mL]. After 72 h, cells were stained using different fluorescently labeled antibodies to check for B-cell proliferation. As shown in Figure 2A, both V3-1 and V3-46s mIgG2a show significantly lower percentages of B-cells in comparison to the BAFF-only and BAFF+IC samples, with percentages very close to the untreated cells. In Figure 2B, it is shown that V3-1 mIgG1 does not reduce the percentage of B-cells significantly from either the BAFF-only or the BAFF+IC, while V3-46s mIgG1 shows a small reduction in B-cell percentage that is significantly lower than BAFF-only and BAFF+IC.

After this assay, we chose to continue with V3-1 mIgG2a, V3-46s mIgG2a, and V3-46s mIgG1 to a dose–response proliferation assay. In this experiment, we expected to evaluate which of the mAbs better reduces B-cell percentages at lower concentrations. We tested both a specific B-cell population (CD21+ CD23+, Follicular B-cells), in Figure 3A, and a general B-cell population in Figure 3B. As shown in both Figure 3A,B, the mAbs reduced the percentage of B-cells in a dose-dependent manner. While in both populations, at the higher concentrations, V3-46s mIgG2a is the antibody that has the greater reduction. Interestingly, in Figure 3B, it is also shown that the mIgG1 isotype control causes a reduction in B-cells percentage, in comparison to the BAFF-only treatment.

At the conclusion of these experiments, we decided to continue in vivo studies with the highest-affinity antibody: V3-46s mIgG2a.

### 3.4. In Vivo Evaluation of V3-46s mIgG2a mAb Using the CIA Model

An in vivo experiment was conducted to evaluate V3-46s mIgG2a antibody’s therapeutic abilities. The experimental design is illustrated in Figure 4A. Mice that were injected with CFA+ collagen are labeled as RA mice, and mice that were injected with CFA+ PBS are labeled as control mice. Based on effective antibody doses that were used to treat mice models in [24] and with BAFF-Trap in [15], RA mice were treated with 6 mg/kg of V3-46s mIgG2a, the appropriate isotype control, and PBS as an additional control, by five intraperitoneal injections. The mice’s arthritic scores were monitored daily, and on day 32, the experiment ended. As shown in Figure 4B, the control mice had an arthritis incidence of 0% throughout the entire experimental time, the RA+PBS mice group reached 100% arthritis incidence by day 28, the RA+IC mice group reached 100% arthritis incidence by day 25, and the RA+anti-BR3 mice group reached a maximal arthritis incidence of 88% by day 32 (Figure 4B). These results suggest that treatment with V3-46s mIgG2a delays the development of arthritis compared to the controls.

The arthritis score of each group is presented in Figure 5A. As shown, there was a significant increase in arthritis score in the RA+IC mice group, with a mean of 8.4 at day 32 in comparison to the control mice group with a mean of 0 throughout the entire experiment, and to the RA+anti-BR3 mice group with a mean score of 2 at day 32. The RA+PBS mice group was not significantly different from the other groups, with a mean score of 6.2.

The area under the curve (AUC) of the arthritis score is presented in Figure 5B. The AUC of the RA+IC mice group, with a mean of 49.6, is also significantly larger than the AUC of the control group, with a mean of 0, and lower than that of the RA+anti-BR3 mice group, with a mean AUC of 11.2. The RA+PBS mice group was not significantly different from the other groups, with a mean AUC of 26.4. When dividing the score by the number of affected paws, as shown in Figure 5C,D, the results are even more significant with a mean arthritis score of 3.1 and a mean AUC of 19.2 in the RA+IC mice group, a mean of 0 in both arthritis score and AUC in the control mice group, and a mean arthritis score of 1.1 and mean AUC of 5.6 in the RA+anti-BR3 mice group. Again, the RA+PBS mice group was not significantly different from the other groups, with a mean arthritis score of 2.2 and a mean AUC of 13.4.

As shown in Figure 6A, the spleen index of the RA+IC mice group, with a mean of 10, was significantly increased in comparison to the control mice group, with a mean of 4.3, and the RA+anti-BR3 mice group, with a mean of 7. The PBS-treated mice show a trend of lower arthritic score compared to the IC group; however, this difference is not statistically significant. The mean of the RA+PBS mice group was 9.2, and it differs from the RA+anti-BR3 mice group with a *p* value of 0.06.

These results suggest that, in addition to delaying disease onset, treatment with V3-46s mIgG2a alleviates the severity of arthritis compared to the controls.

Splenocytes were isolated from mice from the different treatment groups, stained for different leukocyte population markers, and analyzed by flow cytometry. We quantified the percentage of total leukocytes using the CD45 cell marker, macrophages were marked as CD11b+, and the B-cell population was marked as B220+. A significant difference was not observed between any of the groups regarding CD11b+ cell populations. However, as shown in Figure 6B, the B-cell population of the RA+anti-BR3 mice group, with a mean of 5.4%, was significantly decreased in comparison to the RA+PBS mice group, with a mean of 16.9%, and the RA+IC mice group, with a mean of 23.3%. The control mice group was not significantly different from the other treatment groups, with a mean of 18.7%. These results suggest that treatment with V3-46s mIgG2a reduces the percentage of B-cells in the spleen compared to the controls.

The levels of anti-collagen antibodies in the mice sera were evaluated by ELISA. As shown in Figure 6C,D, the levels of anti-collagen antibodies in the sera of all RA mice groups were significantly increased in comparison to the control mice (the control group had no detectable anti-collagen antibodies). However, in the RA+anti-BR3 mice group, there was a significant decrease in anti-collagen antibodies in the sera, with an AUC mean of 0.0025 in comparison to the RA+PBS mice group, with a mean of 0.0037, and the RA+IC mice group, with a mean of 0.0041. Interestingly, we can still see a significant difference between the groups in the lowest serum concentration, with a mean of 0.16 in the RA+PBS mice group, a mean of 0.24 in the RA+IC mice group, and a mean of 0.06 in the RA+anti-BR3 mice group. These results suggest that treatment with V3-46s mIgG2a reduces the concentration of anti-collagen antibodies in the sera compared to the controls.

As for histological analysis, as shown in Figure 7, all four measurements of histological analysis and the total histology score were significantly increased in the RA+PBS mice group, with means of 2.45–2.76 and 10.67, respectively, and in the RA+IC mice group, with means of 2.54–2.92 and 9.87, respectively, as opposed to the RA+anti-BR3 mice group, with means of 0.83–0.95 and 3.63, respectively, and the control mice group, with a mean of 0 in all measurements. An image of the histological section of each group, representative of the mean score of each group, is presented in Appendix A. These results suggest that treatment with V3-46s mIgG2a alleviates the severity of inflammation, pannus formation, cartilage damage, and bone resorption compared to the controls.

## 4. Discussion

Rheumatoid arthritis (RA) is a chronic autoimmune disease with systemic implications, and its timely and effective treatment is essential for preventing irreversible joint damage, disability, and comorbidities. While current treatment options, including conventional synthetic DMARDs, biological DMARDs, and targeted synthetic DMARDs, have significantly advanced disease management, they present various limitations, including adverse events, loss of efficacy over time, and high cost [5,6]. Biologic therapies, such as anti-CD20 monoclonal antibodies like Rituximab, offer targeted B-cell depletion but lack selectivity, affecting both pathogenic and non-pathogenic B-cell populations and increasing the risk of infections [31,32]. Some RA patients fail to demonstrate a durable, long-lasting response to Rituximab due to immunogenicity or loss of CD20 expression of the pathological B-cells [33,34]. Therefore, it is being suggested that replacing (or combining) anti-CD20 antibodies with anti-BAFF or anti-BAFF-R antibodies may be beneficial for patients who cease to benefit from anti-CD20 therapy alone [12,35].

Targeting the BAFF pathway has been proposed for two decades and tested as an approach to treating autoimmune disorders, including RA [15,36,37]. Antibodies specific to BR3 were reported already in 2006; they were developed to investigate the consequences of targeting BR3 in murine models and to assess the potential of BR3 antibodies as human therapeutics. Still, targeting BAFF-R per se in RA has not been reported to the best of our knowledge [38,39,40]. The patent from which we retrieved the sequences of the anti-BAFF-R antibodies we evaluated herein [23] was aimed at hematological malignancies and at Systemic Lupus Erythematosus. We hypothesized that targeting BAFF-R, a receptor critical for B-cell survival and preferentially expressed on autoreactive B-cells, may emerge as a promising strategy in RA. Unlike CD20, BAFF-R allows for more selective depletion of pathogenic B-cells while potentially preserving regulatory and naive populations. This therapeutic approach may improve safety profiles and clinical outcomes while minimizing immune suppression [41,42,43].

In this study, we prepared and evaluated four murine anti-BAFF-R monoclonal antibodies comprising two pairs of variable domains (V3-1 and V3-46s), each fused to either a murine IgG1 or IgG2a Fc. The IgG2a isotype was selected to evaluate potential enhancement of effector functions via ADCC and CDC, given its stronger affinity to Fcγ receptors [25]. Antibodies were expressed in Expi293F cells, with IgG2a variants being purified using protein A columns, and IgG1 variants were purified using protein G affinity chromatography. While V3-1 mIgG1 and both IgG2a variants were expressed efficiently, V3-46s mIgG1 expression was suboptimal, limiting its in vivo evaluation.

Binding and functional assays demonstrated that all four antibodies specifically bound to murine BAFF-R, with V3-1 mIgG2a showing the highest apparent affinity and V3-46s mIgG2a showing superior neutralization capacity in a BAFF-induced splenic B-cell proliferation assays. Notably, V3-1 mIgG1 failed to suppress B-cell proliferation, despite its binding capacity, suggesting structural or isotypic limitations. As a result, V3-46s mIgG2a was selected for in vivo evaluation.

The therapeutic potential of V3-46s mIgG2a was assessed using the collagen-induced arthritis (CIA) mouse model, which recapitulates several pathological and immunological hallmarks of human RA, including elevated BAFF levels [21,27,28]. The therapeutic regimen, initiated post-booster injection, significantly delayed disease onset and reduced overall arthritis severity compared to isotype and PBS controls. Treated mice displayed lower arthritis scores, reduced spleen index, decreased anti-collagen antibody titers, and a marked reduction in splenic B-cell populations, indicating effective immunomodulation. Histological evaluation further confirmed diminished joint inflammation and damage in the V3-46s mIgG2a-treated group.

## 5. Conclusions

In conclusion, our findings support the therapeutic utility of the anti-BR3 antibody V3-46s mIgG2a in ameliorating RA symptoms, delaying disease onset, and reducing the severity of RA in a CIA mouse model. This approach demonstrated both immunological efficacy and disease-modifying activity. While at present our study may be regarded as a preliminary, proof-of-concept study, future studies should focus on establishing a dose response, comparing it to established therapies, pharmacokinetics, long-term safety, and potential combination regimens with existing RA therapeutics to fully realize its clinical potential.

## Figures and Tables

**Figure 1 antibodies-14-00090-f001:**
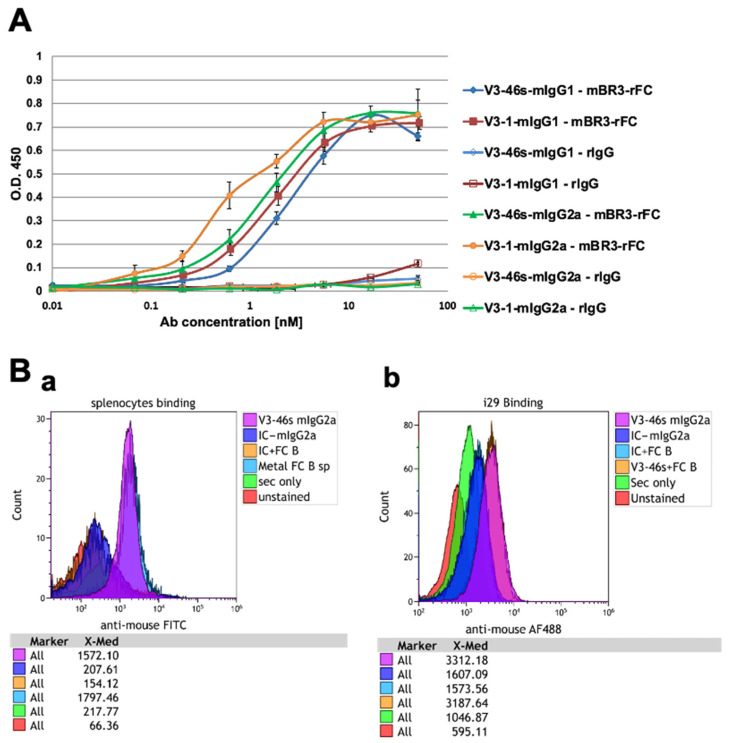
Evaluation of the binding ability of V3-1/V3-46s mIgG1 and V3-1/V3-46s mIgG2a to BR3-rFC and to BR3-expressing cells by ELISA and FACS. (**A**) The binding abilities of all four antibodies were assessed by ELISA. The plate was coated with 5 μg/mL of mBR3-rFC or 5 μg/mL rIgG and blocked with 3% skim milk. 3-fold dilutions starting with 50 nM antibody were added and detected with HRP-conjugated goat anti-mouse H+L antibody. Data are presented as mean ± SD are shown. (**B**) Evaluation of the binding ability of V3-46s mIgG2a to splenocytes/i29 cells by flow cytometry. Histograms of flow cytometry analysis assessing the binding abilities of V3-46s mIgG2a towards BR3 on both splenocytes and i29 cells. The cells were incubated for 30 min at room temperature with the relevant primary mAb (IC/V3-46s mIgG2a) at a concentration of 5 µg/mL. Detection was performed using goat anti-mouse FITC/AF488 as a secondary antibody. Data analysis was conducted using Kaluza software. (**B**(**a**)) V3-46s mIgG2a binding to splenocytes with the presence or absence of FC-blocker (Miltenyi 130-092-575). (**B**(**b**)) V3-46s mIgG2a binding to i29 cells with the presence or absence of Fc-blocker (Miltenyi).

**Figure 2 antibodies-14-00090-f002:**
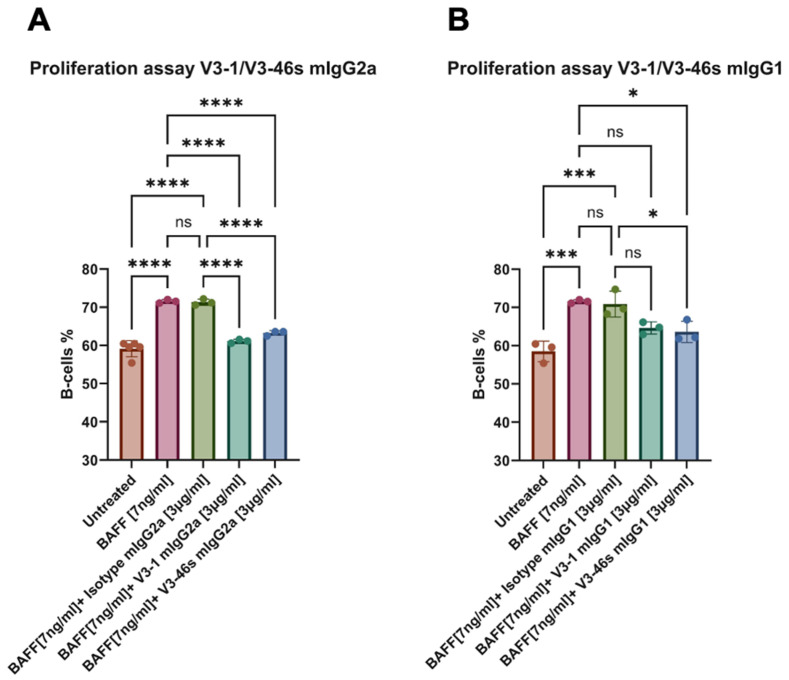
B220+ B-cell proliferation analyzed by flow cytometry. (**A**) mice treated with mouse IgG2a antibodies, (**B**) mice treated with IgG1 antibodies. Splenocytes from C57BL/6 mice (1 × 10^6^ cells/mL) were treated with or without the following: BAFF [7 ng/mL], mIgG2a isotype control mAb (clone-c1.18.4) [3 µg/mL], mIgG1 isotype control mAb (clone- MOPC-21) [3 µg/mL], V3-1 mIgG2a mAb [3 µg/mL], V3-46s mIgG2a mAb [3 µg/mL], V3-1 mIgG1 mAb [3 µg/mL], V3-46s mIgG1 mAb [3 µg/mL], and cultured for 72 h at 37 °C. Cells were then stained for B220 and visualized on a CytoflexL4. * *p* < 0.05, *** *p* < 0.001, **** *p* < 0.0001. ns, without statistical significance (*p* > 0.05).

**Figure 3 antibodies-14-00090-f003:**
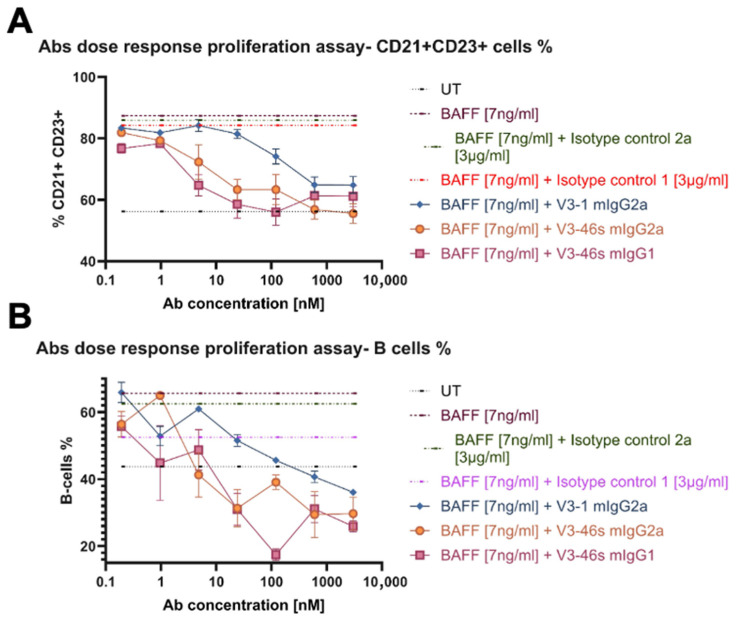
Antibody-mediated dose-dependent inhibition of B220+/CD21+CD23+ B-cell proliferation analyzed by flow cytometry. (**A**) CD21+ CD23+ cells, (**B**) B cells. Splenocytes from C57BL/6 mice (1 × 10^6^ cells/mL) were treated with or without the following: BAFF [7 ng/mL]. mIgG2a isotype control mAb (clone-c1.18.4), mIgG1 isotype control mAb (clone-MOPC-21), V3-1 mIgG2a mAb, V3-46s mIgG2a mAb, and V3-46s mIgG1 mAb were added in a dose-dependent manner, as indicated, with 3-fold dilutions starting at 3 µg/mL, and cultured for 72 h at 37 °C. Cells were then stained for B220, CD21, and CD23, and visualized on a CytoflexL4.

**Figure 4 antibodies-14-00090-f004:**
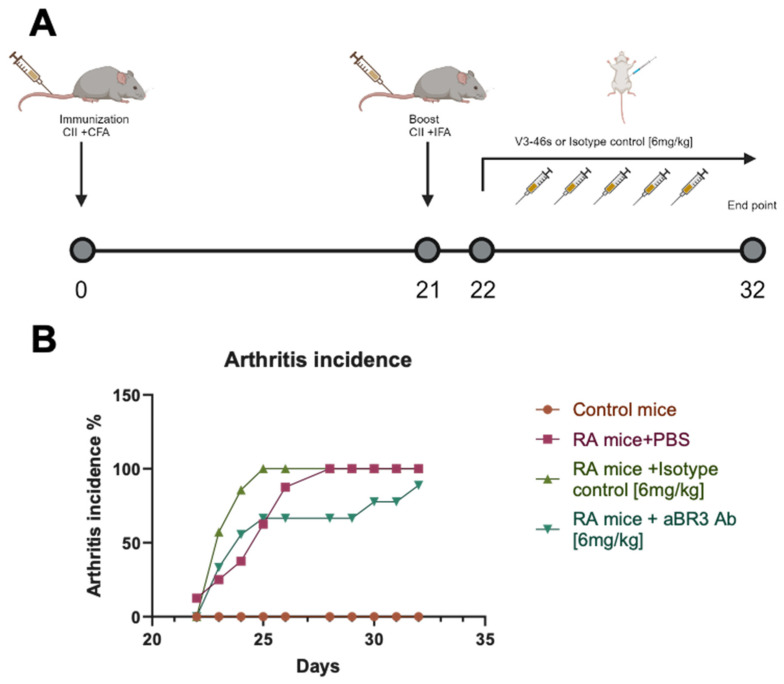
(**A**) Schematic representation of the CIA model. The 8–9-week-old male DBA/1 mice were divided into 4 groups. Arthritis was induced by a primary immunization of bovine type 2 collagen with CFA and a boost immunization of bovine type II collagen with IFA. Anti-BR3 mAb, isotype control, or PBS were injected intraperitoneally (I.P) five times throughout the experiment. On day 32, the mice were euthanized for further analysis. (**B**) Arthritis incidence. The incidence of arthritis score, defined as arthritis score, in at least one paw. Sample sizes: Control mice group (*n* = 2), RA+PBS mice group (*n* = 8), RA+IC mice group (*n* = 7), and RA+anti-BR3 mice group (*n* = 9).

**Figure 5 antibodies-14-00090-f005:**
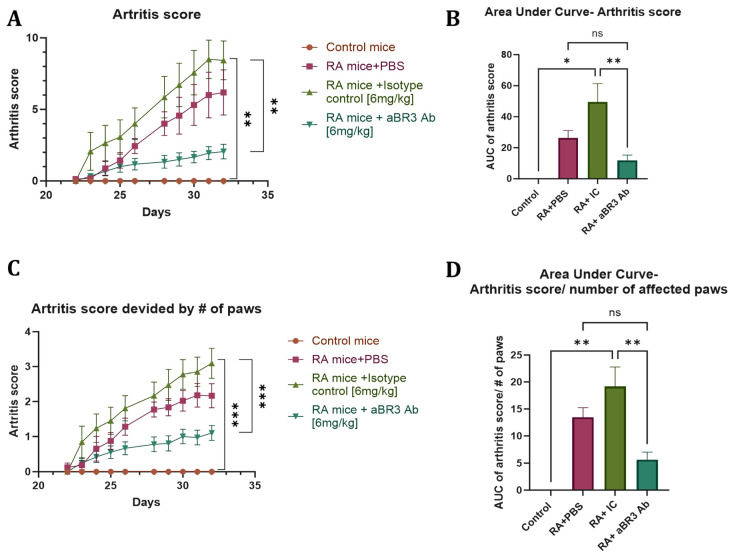
Clinical evaluation of arthritis score. (**A**,**B**) The mice were graded regularly for signs of arthritis. Statistical significance was analyzed by one-way ANOVA using GraphPad Prism 10.2 and is shown as mean ± SEM of both the score at day 32 and the calculated AUC. * *p* < 0.05 and ** *p* < 0.01. ns, without statistical significance (*p* > 0.05). (**C**,**D**) The total score was divided by the number of affected paws in each mouse. Statistical significance was analyzed by one-way ANOVA using GraphPad Prism 10.2 and is shown as mean ± SEM of both the score at day 32 and the calculated AUC. ** *p* < 0.01, *** *p* < 0.001. ns, without statistical significance (*p* > 0.05). Sample sizes: Control mice group (*n* = 2), RA+PBS mice group (*n* = 8), RA+IC mice group (*n* = 7), and RA+anti-BR3 mice group (*n* = 9).

**Figure 6 antibodies-14-00090-f006:**
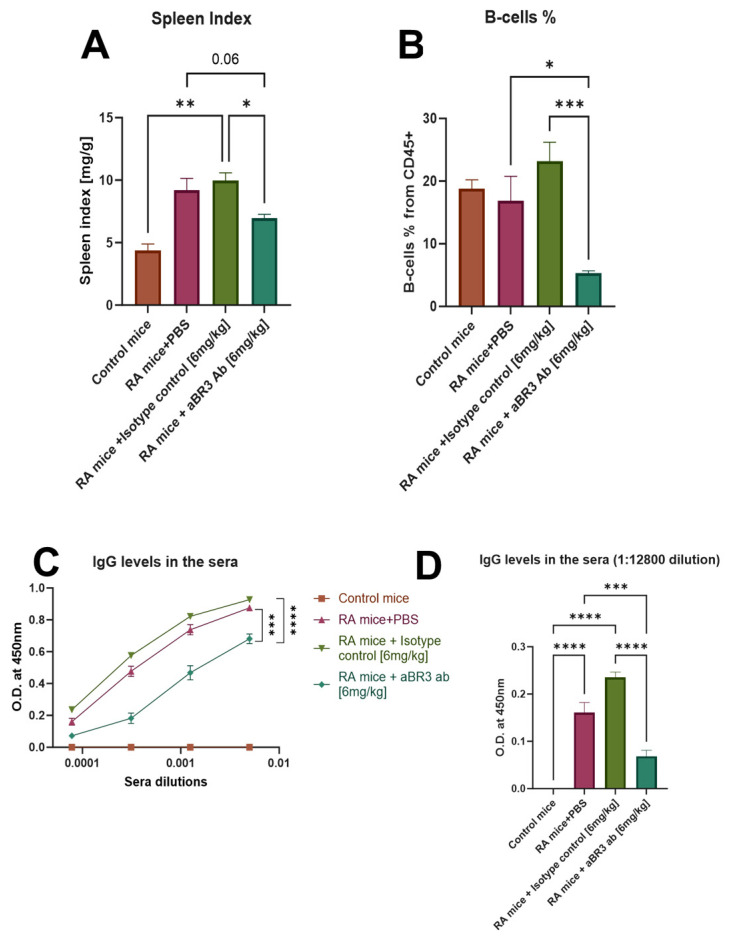
(**A**) Spleen index. Spleen weight [mg]/body weight [g] ratio. Increased ratio of spleen weight/total body weight in CIA mice is consistent with spleen enlargement. Statistical significance was analyzed by one-way ANOVA using GraphPad Prism 10.2 and is shown as mean ± SEM. * *p* < 0.05, ** *p* < 0.01. (**B**) B-cell population in the spleen. Single-cell suspensions from splenocytes were stained using conjugated antibodies. B-cells were defined as CD45+B220+. Statistical significance was analyzed by one-way ANOVA using GraphPad Prism 10.2 and is shown as mean ± SEM. * *p* < 0.05, *** *p* < 0.001. (**C**,**D**) Anti-Collagen antibodies in the sera. The levels of anti-collagen antibodies in the mice sera were evaluated by ELISA. Data was analyzed by one-way ANOVA using GraphPad Prism 10.2 and is shown as mean ± SEM. *** *p* < 0.001, **** *p* < 0.0001. Sample sizes: Control mice group (*n* = 2), RA+PBS mice group (*n* = 8), RA+IC mice group (*n* = 7), and RA+anti-BR3 mice group (*n* = 9).

**Figure 7 antibodies-14-00090-f007:**
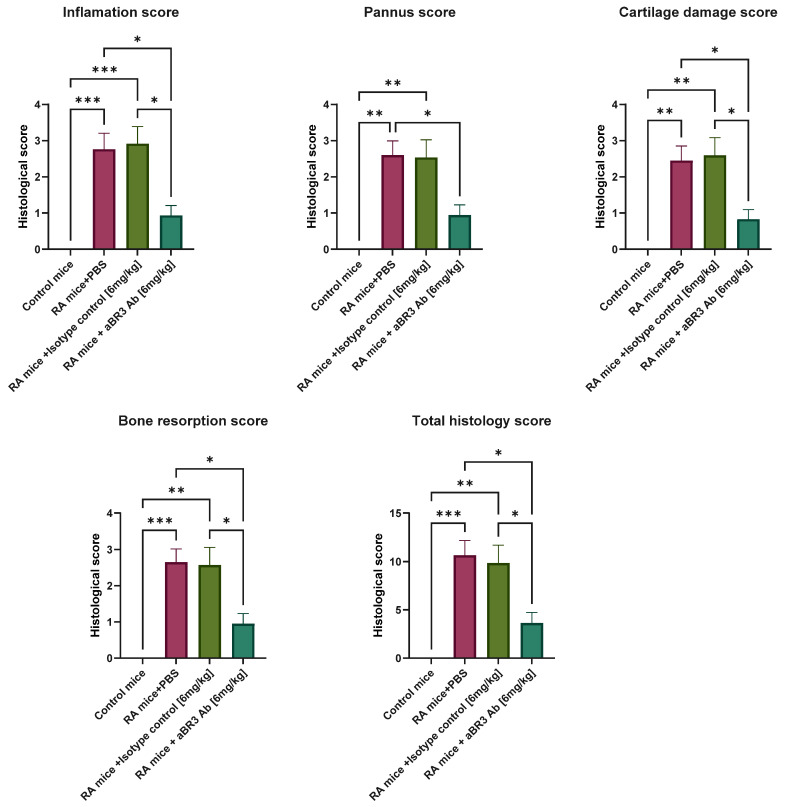
Histological evaluation of the joints. Bar graphs depict the mean (±SEM) for the histological scores, with separate scores for inflammation, pannus formation, bone resorption, cartilage damage, and total histology score; the scores were divided by the ratio of affected joints/number of joints on the slide. Statistical significance was analyzed by one-way ANOVA using GraphPad Prism 10.2 and is shown as mean ± SEM. * *p* < 0.05, ** *p* < 0.01, and *** *p* < 0.001. Sample sizes: Control mice group (*n* = 2), RA+PBS mice group (*n* = 8), RA+IC mice group (*n* = 7), and RA+anti-BR3 mice group (*n* = 9).

**Table 1 antibodies-14-00090-t001:** CIA scoring method (adapted from Hooke Laboratories) (https://hookelabs.com/services/cro/cia.html) (accessed on 1 September 2025).

Paw Score	Clinical Observations
0	Normal paw. No obvious differences in appearance vs. healthy mice.
1	One or two toes inflamed and swollen. No apparent swelling of paw or ankle.
2	Three or more toes inflamed and swollen, but no paw swelling, OR Mild swelling of entire paw.
3	Swelling of entire paw and some toes.
4	Severe swelling of entire paw and all toes.

## Data Availability

The data presented in this study are available on request from the corresponding author.

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
