# Peer review of "Evaluating the Therapeutic Efficacy of an Anti-BAFF Receptor Antibody Using a Rheumatoid Arthritis Mouse Model"

_2073-4468, 2025, doi:10.3390/antib14040090_

Round 1
Reviewer 1 Report
Comments and Suggestions for Authors
In this manuscript Aharon et al. evaluate the potential use of V3-46s mIgG2a, an anti-BAFF-R antibody in a mouse model of rheumatoid arthritis. Initially, four antibody variants were expressed, purified and assessed concerning their binding and neutralizing activity in vitro, and one (the V3-46s mIgG2a) was selected for in vivo evaluation in a collagen-induced arthritis (CIA) model.
Some points should be addressed by the authors:
Authors claim that “BAFF-R (B-cell activating factor receptor) is more selectively expressed on transitional and mature B cells, with elevated expression on autoreactive B cell subsets implicated in autoimmune diseases”. Nevertheless, as also acknowledged by the authors, BAFF “is essential for B cell maturation and survival via its interaction with BAFF-R”. Thus, I wonder how can one justify the targeting of BAFF-R as “a more specific modulation of the B cell compartment” or even claim that such approach could “spare beneficial B cell populations while reducing pathogenic ones.” I believe that this point deserves a better discussion.
Considering the in vivo experiment, Figure 7 clearly shows that “anti BR3 mice group reached a maximal arthritis inciddressed by the authors:ence of 88% in day 32” but that the incidence was increasing (it seems clear that it did not reach 100% because the experiment was interrupted). Considering that, I wonder if 32 days is an appropriate interval for assessing the effects? Please comment on that.
The same question applies when taking into account the data concerning the arthritis score. I understand that at day 32 the RA mice + aBR3 Ab had a lower arthritis score, but would such difference remains since it was in an ascending curve?
If authors want to claim that “These results suggest that treatment with V3-46s mIgG2a alleviates the severity of arthritis compared to the controls.”, a longer follow up would be necessary as evidence to differentiate from alleviated severity from just a delayed observation? The difference is quite important considering the authors discussion about potential therapeutics use.
It is true that the therapeutic potential of V3-46s mIgG2a was assessed using the collagen-induced 441 arthritis (CIA) mouse model, but antibodies were injected starting at day 22, when (according to Figure 7) arthritis incidence was below 50%. Such a protocol is not feasible in a therapeutic approach, where disease (and therefore inflammation) is already, established. Can the authors comment on that?
Finally, and quite curiously, RA mice+PBS show a slower dynamics concerning arthritis incidence (Figure 7). I would like to hear the authors opinion concerning this point.
Minor points
Considering the antibodies used, it would be better to mention, for example, “anti-CD11b-APC-Vio770” instead of just “CD11b-APC-Vio770”.
Figure 11 should include scale bars. Also, information is truncated at this figure, please correct.
Comments on the Quality of English LanguageThe manuscript will benefit from a review of the English-language usage and grammar.
Author Response
Reviewer 1
In this manuscript Aharon et al. evaluate the potential use of V3-46s mIgG2a, an anti-BAFF-R antibody in a mouse model of rheumatoid arthritis. Initially, four antibody variants were expressed, purified and assessed concerning their binding and neutralizing activity in vitro, and one (the V3-46s mIgG2a) was selected for in vivo evaluation in a collagen-induced arthritis (CIA) model.
Some points should be addressed by the authors:
Authors claim that “BAFF-R (B-cell activating factor receptor) is more selectively expressed on transitional and mature B cells, with elevated expression on autoreactive B cell subsets implicated in autoimmune diseases”. Nevertheless, as also acknowledged by the authors, BAFF “is essential for B cell maturation and survival via its interaction with BAFF-R”. Thus, I wonder how can one justify the targeting of BAFF-R as “a more specific modulation of the B cell compartment” or even claim that such approach could “spare beneficial B cell populations while reducing pathogenic ones.” I believe that this point deserves a better discussion.
Considering the in vivo experiment, Figure 7 clearly shows that “anti BR3 mice group reached a maximal arthritis inciddressed by the authors:ence of 88% in day 32” but that the incidence was increasing (it seems clear that it did not reach 100% because the experiment was interrupted). Considering that, I wonder if 32 days is an appropriate interval for assessing the effects? Please comment on that.
The same question applies when taking into account the data concerning the arthritis score. I understand that at day 32 the RA mice + aBR3 Ab had a lower arthritis score, but would such difference remains since it was in an ascending curve?
If authors want to claim that “These results suggest that treatment with V3-46s mIgG2a alleviates the severity of arthritis compared to the controls.”, a longer follow up would be necessary as evidence to differentiate from alleviated severity from just a delayed observation? The difference is quite important considering the authors discussion about potential therapeutics use.
It is true that the therapeutic potential of V3-46s mIgG2a was assessed using the collagen-induced 441 arthritis (CIA) mouse model, but antibodies were injected starting at day 22, when (according to Figure 7) arthritis incidence was below 50%. Such a protocol is not feasible in a therapeutic approach, where disease (and therefore inflammation) is already, established. Can the authors comment on that?
Finally, and quite curiously, RA mice+PBS show a slower dynamics concerning arthritis incidence (Figure 7). I would like to hear the authors opinion concerning this point.
Minor points
Considering the antibodies used, it would be better to mention, for example, “anti-CD11b-APC-Vio770” instead of just “CD11b-APC-Vio770”.
Figure 11 should include scale bars. Also, information is truncated at this figure, please correct.
Comments on the Quality of English Language
The manuscript will benefit from a review of the English-language usage and grammar.
Submission Date
15 September 2025
Date of this review
03 Oct 2025 20:17:43
Responses:
Some points should be addressed by the authors:
- Authors claim that “BAFF-R (B-cell activating factor receptor) is more selectively expressed on transitional and mature B cells, with elevated expression on autoreactive B cell subsets implicated in autoimmune diseases”. Nevertheless, as also acknowledged by the authors, BAFF “is essential for B cell maturation and survival via its interaction with BAFF-R”. Thus, I wonder how can one justify the targeting of BAFF-R as “a more specific modulation of the B cell compartment” or even claim that such approach could “spare beneficial B cell populations while reducing pathogenic ones.” I believe that this point deserves a better discussion.
Response: we provide an explanation for that in the introduction of the revised manuscript, including newly added references, on Page 3 lines 70-79 of the numbered version of our revised manuscript.
- Considering the in vivo experiment, Figure 7 clearly shows that “anti BR3 mice group reached a maximal arthritis inciddressed by the authors:ence of 88% in day 32” but that the incidence was increasing (it seems clear that it did not reach 100% because the experiment was interrupted). Considering that, I wonder if 32 days is an appropriate interval for assessing the effects? Please comment on that.
Response: we provide an explanation for that starting on Page 6 line 185 of the numbered version of our revised manuscript. We would like to comment that with this RA mouse model, cures are rarely achieved and delay of onset or reduction in the severity of arthritic score, are considered evidence for efficacy.
- The same question applies when taking into account the data concerning the arthritis score. I understand that at day 32 the RA mice + aBR3 Ab had a lower arthritis score, but would such difference remains since it was in an ascending curve?
Response: The same response as for comment 2. We would like to comment that with this RA mouse model, observing a an ascending curve (of the treatment group) in the later time point of treatment is a common observation.
- If authors want to claim that “These results suggest that treatment with V3-46s mIgG2a alleviates the severity of arthritis compared to the controls.”, a longer follow up would be necessary as evidence to differentiate from alleviated severity from just a delayed observation? The difference is quite important considering the authors discussion about potential therapeutics use.
Response: following our response to comments 2 and 3, suggesting a therapeutic potential based on statistically-significant differences between the treatment to the control group is acceptable with this RA mouse model.
- It is true that the therapeutic potential of V3-46s mIgG2a was assessed using the collagen-induced 441 arthritis (CIA) mouse model, but antibodies were injected starting at day 22, when (according to Figure 7) arthritis incidence was below 50%. Such a protocol is not feasible in a therapeutic approach, where disease (and therefore inflammation) is already, established. Can the authors comment on that?
Response: The reviewer is right in commenting that a such a therapeutic approach is not directly applicable to human patients. However, this RA mouse model was used by us to establish, for the first time, that an antibody that inhibits BR3 may be efficacious is an RA mouse model. We believe our results suggest that it is. We provide additional references for the use of this mouse RA model, as can be seen on Page 6 lines 171-172 of the numbered version of our revised manuscript.
- Finally, and quite curiously, RA mice+PBS show a slower dynamics concerning arthritis incidence (Figure 7). I would like to hear the authors opinion concerning this point.
Response: The reviewer is right in commenting that this trend can be seen and what now is Figure 4 (seen also in Figures 5 and 6). However, as we state on Page 15 lines 368-370 of the numbered version of our revised manuscript, this difference is not statistically significant. It is now rare to observe in antibody treatments of mice that the injection of an isotype control antibody differs from injecting just buffer, and this may be such an instance.
Minor points
- Considering the antibodies used, it would be better to mention, for example, “anti-CD11b-APC-Vio770” instead of just “CD11b-APC-Vio770”.
Response: We made the change as recommended by the reviewer. See page 8 lines 208-210 of the numbered version of our revised manuscript.
- Figure 11 should include scale bars. Also, information is truncated at this figure, please correct.
Response: The figure was corrected as suggested by the reviewer. It is now Supplementary Figure 2 on page 27 of the numbered version of our revised manuscript.
As for scale bars: as shown in the legend of Supplementary Figure 2, )on page 27 lines 687-691 of the numbered version of our revised manuscript), the sections shown are of whole paws at X10 magnification. In such figures, scale bars are usually not shown. See for example the WEB site of the CRO that provide RA mice for experimentation: https://hookelabs.com/services/cro/cia.html
- Comments on the Quality of English Language
The manuscript will benefit from a review of the English-language usage and grammar.
Response: As part of the major revision, we carried out according to the recommendations of the reviewer, we also thoroughly edited the manuscript for language and grammar. We hope they are not satisfactory.
Reviewer 2 Report
Comments and Suggestions for Authors
Review report – Antibodies
Manuscript:
Evaluating the Therapeutic Efficacy of an Anti-BAFF Receptor Antibody Using a Rheumatoid Arthritis Mouse Model
(Aharon et al.)
General comment
This manuscript explores the therapeutic potential of a BAFF-R–targeting antibody in rheumatoid arthritis (RA), combining in vitro characterization with an in vivo evaluation in the collagen-induced arthritis (CIA) model. The rationale is relevant, as BAFF-R may represent a more selective target than CD20. The experiments are coherent, and the data suggest that V3-46s mIgG2a can delay disease onset, reduce severity, and modulate B-cell populations. However, the manuscript has several important weaknesses. The in vivo study relies on a single treatment protocol, with no dose-response analysis or comparison to established therapies. No validation with human samples is provided, limiting translational conclusions. Figures are overly numerous (12) and of insufficient visual quality, and the discussion is underdeveloped, with very few references (~19 in total, whereas at least ~40 would be expected). The conclusions currently overstate the significance of the findings, which should be framed as preliminary, proof-of-concept data. I recommend major revisions.
Major comments
- Experimental design limitations
- The in vivo evaluation is restricted to a single antibody (V3-46s mIgG2a), one dosing regimen.
- No dose-response or therapeutic comparison (e.g. anti-CD20, anti-BAFF, anti-TNF) is included, which limits interpretation of efficacy.
- The absence of validation on human samples (ex vivo PBMCs or patient data) limits translational claims.
- Risk of overinterpretation
- The data demonstrate efficacy in CIA, but claims of “therapeutic utility” and “better safety profile” are speculative.
- Translational advantages of BAFF-R targeting must be presented as hypotheses, not conclusions.
- The CIA model has limitations in recapitulating RA pathogenesis, which should be explicitly acknowledged.
- Figures
- The number of figures (12) is excessive. A reasonable number would be ~7 main figures. Reducing the overall number of figures will also enhance clarity and readability of the manuscript.
- Several datasets (e.g. antibody binding and proliferation assays) should be combined into multi-panel figures.
- Minor or confirmatory results can be moved to supplementary material.
- The overall quality of the figures is not acceptable; they should be homogeneous, of high resolution, and must not rely on screenshots. Figures should be generated with appropriate scientific software (e.g., GraphPad Prism) rather than basic tools such as Excel.
- Figure legends need to be substantially revised. They should be more complete and include all necessary information (e.g., sample size, statistical tests, definitions of symbols and abbreviations). Each legend should allow the figure to be understood independently, without requiring the main text for interpretation.
- References and discussion
- The manuscript cites only 19 references, which is insufficient. At least ~40 should be included to provide adequate background.
- The discussion should be expanded, with critical comparisons to existing B-cell–targeting therapies (rituximab, belimumab) and cytokine inhibitors (anti-TNF, anti-IL6).
- The novelty of BAFF-R targeting should be contextualized more clearly against prior literature.
- Several key references in the field are missing, which weakens the contextualization of the results and limits the depth of the discussion.
- Statistical analysis
- Group sizes (n=7–9) are appropriate for CIA experiments. However, the statistical methods need clearer justification (choice of tests, correction for multiple comparisons).
- Exact sample sizes should be included in figure legends.
Minor comments
- The English language should be edited for grammar and clarity.
- A schematic or table summarizing BAFF/BAFF-R–targeted strategies in autoimmune diseases would add useful context.
- Reference formatting should be checked for consistency.
Conclusion
This manuscript provides interesting proof-of-concept evidence that BAFF-R–targeted antibodies can attenuate disease in a CIA model. The results are promising but remain preliminary, and the manuscript requires substantial revisions. In particular, the figures and legends should be consolidated and improved for clarity, the quality is not acceptable in its current form. The discussion needs to be expanded and supported by a broader set of references, the experimental limitations must be more explicitly acknowledged, and the translational claims should be moderated to reflect the exploratory nature of the work. I recommend major revisions.
No anomalies were detected regarding compliance with Ethical Guidelines and ARRIVE recommendations in relation to the animal experiments described in the manuscript.

The English language should be edited for grammar and clarity.
Author Response
Reviewer 2
This manuscript explores the therapeutic potential of a BAFF-R–targeting antibody in rheumatoid arthritis (RA), combining in vitro characterization with an in vivo evaluation in the collagen-induced arthritis (CIA) model. The rationale is relevant, as BAFF-R may represent a more selective target than CD20. The experiments are coherent, and the data suggest that V3-46s mIgG2a can delay disease onset, reduce severity, and modulate B-cell populations.
However, the manuscript has several important weaknesses. The in vivo study relies on a single treatment protocol, with no dose-response analysis or comparison to established therapies. No validation with human samples is provided, limiting translational conclusions.
Figures are overly numerous (12) and of insufficient visual quality, and the discussion is underdeveloped, with very few references (~19 in total, whereas at least ~40 would be expected). The conclusions currently overstate the significance of the findings, which should be framed as preliminary, proof-of-concept data. I recommend major revisions.
Major comments
Experimental design limitations
- The in vivo evaluation is restricted to a single antibody (V3-46s mIgG2a), one dosing regimen.
- No dose-response or therapeutic comparison (e.g. anti-CD20, anti-BAFF, anti-TNF) is included, which limits interpretation of efficacy.
- The absence of validation on human samples (ex vivo PBMCs or patient data) limits translational claims.
Risk of overinterpretation
- The data demonstrate efficacy in CIA, but claims of “therapeutic utility” and “better safety profile” are speculative.
- Translational advantages of BAFF-R targeting must be presented as hypotheses, not conclusions.
- The CIA model has limitations in recapitulating RA pathogenesis, which should be explicitly acknowledged.
Figures
- The number of figures (12) is excessive. A reasonable number would be ~7 main figures. Reducing the overall number of figures will also enhance clarity and readability of the manuscript.
- Several datasets (e.g. antibody binding and proliferation assays) should be combined into multi-panel figures.
- Minor or confirmatory results can be moved to supplementary material.
- The overall quality of the figures is not acceptable; they should be homogeneous, of high resolution, and must not rely on screenshots. Figures should be generated with appropriate scientific software (e.g., GraphPad Prism) rather than basic tools such as Excel.
- Figure legends need to be substantially revised. They should be more complete and include all necessary information (e.g., sample size, statistical tests, definitions of symbols and abbreviations). Each legend should allow the figure to be understood independently, without requiring the main text for interpretation.
References and discussion
- The manuscript cites only 19 references, which is insufficient. At least ~40 should be included to provide adequate background.
- The discussion should be expanded, with critical comparisons to existing B-cell–targeting therapies (rituximab, belimumab) and cytokine inhibitors (anti-TNF, anti-IL6).
- The novelty of BAFF-R targeting should be contextualized more clearly against prior literature.
- Several key references in the field are missing, which weakens the contextualization of the results and limits the depth of the discussion.
Statistical analysis
- Group sizes (n=7–9) are appropriate for CIA experiments. However, the statistical methods need clearer justification (choice of tests, correction for multiple comparisons).
- Exact sample sizes should be included in figure legends.
Minor comments
The English language should be edited for grammar and clarity.
A schematic or table summarizing BAFF/BAFF-R–targeted strategies in autoimmune diseases would add useful context.
Reference formatting should be checked for consistency.
Conclusion
This manuscript provides interesting proof-of-concept evidence that BAFF-R–targeted antibodies can attenuate disease in a CIA model. The results are promising but remain preliminary, and the manuscript requires substantial revisions. In particular, the figures and legends should be consolidated and improved for clarity, the quality is not acceptable in its current form. The discussion needs to be expanded and supported by a broader set of references, the experimental limitations must be more explicitly acknowledged, and the translational claims should be moderated to reflect the exploratory nature of the work. I recommend major revisions.
No anomalies were detected regarding compliance with Ethical Guidelines and ARRIVE recommendations in relation to the animal experiments described in the manuscript.
Comments on the Quality of English Language
The English language should be edited for grammar and clarity.
Submission Date
15 September 2025
Date of this review
29 Sep 2025 10:11:01
Responses:
Reviewer 2
This manuscript explores the therapeutic potential of a BAFF-R–targeting antibody in rheumatoid arthritis (RA), combining in vitro characterization with an in vivo evaluation in the collagen-induced arthritis (CIA) model. The rationale is relevant, as BAFF-R may represent a more selective target than CD20. The experiments are coherent, and the data suggest that V3-46s mIgG2a can delay disease onset, reduce severity, and modulate B-cell populations.
- However, the manuscript has several important weaknesses. The in vivo study relies on a single treatment protocol, with no dose-response analysis or comparison to established therapies. No validation with human samples is provided, limiting translational conclusions.
Response: The reviewer is right in commenting that without a dose response or comparison to other therapies (such as anti CD20), our results are not directly applicable to human patients. However, this study with this RA mouse model was used by us to establish, for the first time, that an antibody that inhibits BR3 may be efficacious is an RA mouse model. We believe our results suggest that it is. As for human samples, testing such is beyond the scope of our manuscript, which is focused on a mouse study. We do provide additional references pertaining to the relevance of the BAFF / BAFF-R pathway to autoimmunity and to RA. See for examples on page 3 lines 71-80 of the numbered version of our revised manuscript.
- Figures are overly numerous (12) and of insufficient visual quality, and the discussion is underdeveloped, with very few references (~19 in total, whereas at least ~40 would be expected). The conclusions currently overstate the significance of the findings, which should be framed as preliminary, proof-of-concept data. I recommend major revisions.
Response: As suggested by the reviewer, we carried out the following changes:
- We improved the quality of the Figures.
- We further developed the Discussion, see page 20 line 455 to page 21 line 504 of the numbered version of our revised manuscript.
- Along with the revision, we added more references, most in the introduction and the discussion sections, The revised manuscript now includes 43 references. See page 22 line 526 to page 25 line 666 of the numbered version of our revised manuscript.
- We modified the conclusion paragraph, stating that our study is a preliminary, proof-of-concept study. See page 21 lines 505-511 of the numbered version of our revised manuscript.
- Major comments
Experimental design limitations
- The in vivo evaluation is restricted to a single antibody (V3-46s mIgG2a), one dosing regimen.
- No dose-response or therapeutic comparison (e.g. anti-CD20, anti-BAFF, anti-TNF) is included, which limits interpretation of efficacy.
- The absence of validation on human samples (ex vivo PBMCs or patient data) limits translational claims.
Risk of overinterpretation
- The data demonstrate efficacy in CIA, but claims of “therapeutic utility” and “better safety profile” are speculative.
- Translational advantages of BAFF-R targeting must be presented as hypotheses, not conclusions.
Response: we respond to these comments above in our response to query 10.
- The CIA model has limitations in recapitulating RA pathogenesis, which should be explicitly acknowledged.
Response: without going into a lengthy discussion of the general limitations of using rodent experiment to study human pathologies, we respectfully argue that, the CIA mouse model we used is perhaps the RA mouse model most closely representing the human condition, including the relevance of the BAFF pathway in RA. We write about it and provide relevant references on page 6 lines 171-172 of the numbered version of our revised manuscript.
- Figures
- The number of figures (12) is excessive. A reasonable number would be ~7 main figures. Reducing the overall number of figures will also enhance clarity and readability of the manuscript.
Response: As suggested by the reviewer, we moved two figures to the supplement and combined other figures to multi-panel figures. In the revised manuscript we have 7 figures in the main text and two supplemental figures.
- Several datasets (e.g. antibody binding and proliferation assays) should be combined into multi-panel figures.
Response: This was carried out. see our response to query 14.
- Minor or confirmatory results can be moved to supplementary material.
Response: This was carried out. see our response to query 14. Specifically, Supplemental Figures 1 and 2 that in the original submission were Figures 1 and 11.
- The overall quality of the figures is not acceptable; they should be homogeneous, of high resolution, and must not rely on screenshots. Figures should be generated with appropriate scientific software (e.g., GraphPad Prism) rather than basic tools such as Excel.
Response: We improved the quality of the Figures as suggested.
- Figure legends need to be substantially revised. They should be more complete and include all necessary information (e.g., sample size, statistical tests, definitions of symbols and abbreviations). Each legend should allow the figure to be understood independently, without requiring the main text for interpretation.
Response: The figure legends were improved as suggested by the reviewer.
- References and discussion
- The manuscript cites only 19 references, which is insufficient. At least ~40 should be included to provide adequate background.
Response: As recommended by the reviewer, we added more references, most in the introduction and the discussion sections, The revised manuscript now includes 43 references. See page 22 line 526 to page 25 line 666 of the numbered version of our revised manuscript.
- The discussion should be expanded, with critical comparisons to existing B-cell–targeting therapies (rituximab, belimumab) and cytokine inhibitors (anti-TNF, anti-IL6).
- The novelty of BAFF-R targeting should be contextualized more clearly against prior literature.
Response: we expanded the discussion as suggested by the reviewer (see page 20 line 455 to page 21 line 512 of the numbered version of our revised manuscript) and provided a more comprehensive introduction to RA including to existing therapeutic approaches (see page 3 lines 71 80 of the numbered version of our revised manuscript). We hope that the requested comparisons to existing B-cell targeting therapies are provided in our revised manuscript. Of-note, we do not claim that targeting the BAFF / BAFF-R axis in RA is novel. On the contrary, we mention it was already proposed in the literature (see page 20 lines 76-80 of the numbered version of our revised manuscript). What we do argue is that a BR3-neutralizing antibody was not evaluated in a mouse RA model before (see page 20 lines 76-80 of the numbered version of our revised manuscript).
- Several key references in the field are missing, which weakens the contextualization of the results and limits the depth of the discussion.
Response: we added more references to address this comment. See our response to query 19.
- Statistical analysis
- Group sizes (n=7–9) are appropriate for CIA experiments. However, the statistical methods need clearer justification (choice of tests, correction for multiple comparisons).
- Exact sample sizes should be included in figure legends.
Response: we provide the additional information in the legends of the relevant figures (Figures 4-6 of the revised manuscript). We justify the choice of the statistical approach on page 8 lines 224-230 of the numbered version of our revised manuscript.
- Minor comments
The English language should be edited for grammar and clarity.
Response: As part of the major revision, we carried out according to the recommendations of the reviewer, we also thoroughly edited the manuscript for language and grammar. We hope they are not satisfactory.
- A schematic or table summarizing BAFF/BAFF-R–targeted strategies in autoimmune diseases would add useful context.
Response: we refer the reader to an excellent recent review where such a Table is shown (Reference 10, References 11 and 12 also provide information about targeting the BAFF / BAFF-R axis in autoimmunity in general. See page 3 lines 70-79 of the numbered version of our revised manuscript.
- Reference formatting should be checked for consistency.
Response: As part of the major revision, we carried out according to the recommendations of the reviewer, we also thoroughly edited the manuscript for language and grammar. We hope they are not satisfactory. The references were generated using Endnote with MDPI style, which is the recommendation by the journal “Antibodies”
- Conclusion
This manuscript provides interesting proof-of-concept evidence that BAFF-R–targeted antibodies can attenuate disease in a CIA model. The results are promising but remain preliminary, and the manuscript requires substantial revisions. In particular, the figures and legends should be consolidated and improved for clarity, the quality is not acceptable in its current form. The discussion needs to be expanded and supported by a broader set of references, the experimental limitations must be more explicitly acknowledged, and the translational claims should be moderated to reflect the exploratory nature of the work. I recommend major revisions.
Response: we thank the reviewer for the useful comments, that, we hope, we responded adequately.
- No anomalies were detected regarding compliance with Ethical Guidelines and ARRIVE recommendations in relation to the animal experiments described in the manuscript.
- The English language should be edited for grammar and clarity.
Response: As part of the major revision, we carried out according to the recommendations of the reviewer, we also thoroughly edited the manuscript for language and grammar. We hope they are not satisfactory.
Round 2
Reviewer 1 Report
Comments and Suggestions for Authors
In the revised version, the authors approached the main questions raised by this reviewer. They better justify BAFF-R targeting in RA treatment, and also added that “BAFF-R targeting may also be combined with anti CD20 antibodies for more effective therapy of RA”, clearly acknowledging limitations of such approach.
As an answer to my main questionings about the experiments, authors argued that they are using a well established and accepted RA model, although they also acknowledge the limitations of such model. They highlight the limitations of the model and the fact that these are preclinical studies. I understand their point and believe that, despite the model limitations, it contributes to the advance on the field.
Reviewer 2 Report
Comments and Suggestions for Authors
The authors have carefully revised their manuscript and have satisfactorily addressed all major and minor comments raised during the initial review. The revised version demonstrates substantial improvements in clarity, figure quality, and scientific framing. The number of figures has been reduced and reorganized for readability, legends are now complete and informative, and the overall presentation meets the journal’s standards.
The Discussion has been significantly expanded and now includes appropriate comparisons with established B-cell–targeting and cytokine-inhibiting therapies, supported by an adequate number of references (43 in total). The authors have also moderated their claims, clearly presenting the work as a preliminary, proof-of-concept study, which is consistent with the data provided.
While the experimental design remains limited to a single treatment protocol without validation on human samples, this is now explicitly acknowledged as a limitation. The manuscript is coherent, well written, and provides a meaningful contribution to the understanding of BAFF-R targeting in experimental arthritis.
I therefore consider that all previous concerns have been appropriately addressed and recommend the manuscript for publication in Antibodies in its current form.